# Healing responses at the angle after micro-invasive glaucoma surgery-an AS-OCT study

**Aparna Rao** *, **Sujoy Mukherjee**

Glaucoma Service, LV Prasad Eye Institute, Hyderabad, India

* aparna@lvpei.org

## Abstract

### Purpose

To evaluate structural alterations and healing responses in the trabecular meshwork region with optical coherence tomography (AS-OCT) following after gonioscopy assisted transluminal trabeculotomy (GATT) and microincisional trabeculectomy (MIT).

### Methods

73 eyes of 67 patients (M:F = 45:22) with $\geq$6 months of follow-up after MIT (n = 41) or GATT (n = 32) with or without combined cataract surgery were included for this prospective study. The angle as seen on AS-OCT at 1, 3, 6 months after surgery were evaluated for structural alterations like peripheral anterior synechiae (PAS), hyphema, and hyperreflective scarring responses. The scarring was graded according to the linear extent measured from the centre of the trabecular meshwork (TM) gutter to the sclera/cornea as mild (<250μ), moderate (250–500μ), and severe(>500μ), while the pattern of scarring was graded as open saucer/gutter, closed gutter, and trench pattern. The association of the need for medication or surgical outcome and clinical variables and AS-OCT parameters including the pattern and severity of scarring were analysed using multivariate regression.

### Results

All eyes achieved significant reduction of IOP and number of medications with a final IOP of 15±3.2mm Hg at a mean follow-up of 8±32. months. While mild scarring was seen more common in MIT, severe scarring was seen in >65% of GATT eyes compared to 31% of MIT eye, p<0.001. An open saucer was equally seen in MIT and GATT while the trench pattern was more commonly seen in GATT eyes (>50%). Severe scarring in a trench pattern seemed to predict the need for medications for IOP control, though they independently did not seem to influence the final IOP or surgical outcome.

### Conclusion

A severe form of scarring in a trench pattern on AS-OCT predicted the need for glaucoma medications after MIGS surgery. Regular monitoring of the scarring responses by AS-OCT and clinical examination are necessary to identify those at need for medications after MIGS.

**Data Availability Statement:** All relevant data are within the paper.

**Funding:** The author(s) received no specific funding for this work.

**Competing interests:** The authors have declared that no competing interests exist.

## Introduction

Microinvasive glaucoma surgery (MIGS) has ushered in a minimally invasive surgical option for intraocular pressure (IOP) control [1–5]. These have reduced surgery time and complication rates with comparable efficacy and safety profiles compared to traditional filtering surgeries [1, 2, 4, 5]. The 2-year success rates of these procedures vary from 50–80% across studies, with varying rates reported for different procedures like gonioscopy assisted transluminal trabeculotomy (GATT), MIT, stents, or bent needle angle goniectomy, BANG, [2–11]. The causes for failure of IOP raise after surgery has been attributed to a variety of reasons including macrohyphema, TM reattachment, peripheral anterior synechiae, excessive fibrotic response in the angle or a hypertrophic scar at the surgical site [6, 7, 9].

The predictors for success after MIGS are unclear with better outcomes reported in patients aged <60 years, and primary open-angle glaucoma [1, 7, 9]. Intraoperative extent of blanching achieved, or the extent/pattern of trypan blue staining are also reported to predict the success rates after microincisional trabeculectomy [5, 6]. Yet, it is understood that any incisional site would undergo wound healing responses as in any other part of the eye. In the angle, such healing responses may be responsible for IOP spikes or failure which counteracts the purpose of MIGS [1, 7, 12, 13]. Surprisingly, the healing responses after MIGS has not been studied till now. Our earlier study identified causes of raised IOP after GATT in the immediate or early postoperative period using AS-OCT [7]. AS-OCT offers a non-invasive method of visualising the angle structures and the temporal healing responses after MIGS [11–14]. This study compares the healing responses of the surgical site and the areas adjacent to the TM region after MIT and GATT.

## Methods

This was a prospective ongoing study comparing outcomes of GATT versus MIT in primary and pseudoexfoliation glaucoma. This was approved by the institutional review board of LV Prasad Eye Institute and adhered to the tenets of declaration of Helsinki. An informed written consent was obtained from all patients before recruitment into the study.

All patients diagnosed as primary glaucoma (Primary open-angle or angle closure, POAG and PACG, respectively or pseudoexfoliation glaucoma, PXG) with uncontrolled IOP>21mm Hg and who underwent MIT or GATT from 11th March 2022-30th October 2023, with >6months follow-up were included. Those who underwent conventional trabeculectomy either as an alternative to MIT, unwilling patients, secondary glaucoma, age<40 years, non-availability of good quality data (clinical or others), or those who required it because of unsuccessful MIT, were excluded from the analysis. Those that required concomitant cataract surgery underwent cataract extraction first followed by MIT or GATT after IOL insertion and formation of the AC with viscoelastic. All surgeries were preferably done through the temporal corneal incision with the TM being incised in the nasal quadrant for MIT and threading of the SC starting firm the nasal quadrant for GATT.

### Surgical procedure

The surgical procedure for MIT is described elsewhere [5]. Briefly, after a goniotomy using MVR blade from the temporal corneal incision after IOL insertion. Now, a straight vitreoretinal scissors is directed perpendicularly to give two radial cuts at the upper trabecular edge along 4–6 clock hours in the nasal quadrant. The TM cut edge is grasped and gentle traction applied using a 25-gauge vitreoretinal end-gripping forceps to strip it away in a single motion.

The AC is now washed thoroughly washed to remove viscoelastic and blood with special focus given to the region of the MIT. While washing, care was given to see the episcleral wave

or blanching seen in the quadrant of MIT and away from the incised site. This was now followed by injection of trypam blue under air and care was taken to see the dye exiting through the veins in the nasal, superior or inferior quadrant.

The procedure for 5–0 prolene suture GATT is described elsewhere in detail [7, 9, 10]. Briefly, a 5–0 prolene is threaded into the Schlemm's canal (SC) after goniotomy with an MVR bade after which it is advanced 360 degrees of the SC using microforceps. The leading edge is grasped when it reaches the other cut end of the goniotomy, and the two ends are pulled in opposite directions to unroof the SC. This is followed by AC wash and pilocarpine injection.

Intraoperative complications like AC bleed, iris trauma or dialysis were noted. Postoperatively, the need for aggressive topical steroids or systemic steroids for intense AC inflammation was also noted. The IOP over 1day, 1month, 3 months and 6months with the need for AGMs were recorded.

The postoperative regimen included topical steroids in tapering doses and antibiotics for a week. Topical anti-glaucoma medications were started if the IOP was not as per the target IOP. Additional surgeries like incisional trabeculectomy were planned in the event of uncontrolled IOP despite medications.

## AS-OCT image acquisition

The angle was imaged using swept-source AS-OCT line scan protocol (Topcon DRI Triton Plus (Version 10.19) that has a 6mm scan length. This captures high-resolution cross-sectional images of the angle structures including the TM and the SC. All scans were done by the same trained and experienced examiner. The scan line was placed in the nasal and temporal quadrants to image the nasal incised site in all eyes undergoing GATT or MIT and only scans of good quality and a signal-noise ratio>40 were included. The same region (in clock hours) was imaged at each visit to ensure comparison over follow-ups. Given the acquisition of multiple scans by the AS-OCT device, only the highest-quality images devoid of motion or artifacts were chosen for the analysis. Scarring in the TM, known as a hypertrophic scar, was identified as a hyperintense region (compared to the corneal reflectivity) in the nasal incised site of the TM which may extend both anteroposteriorly underneath the incised region in the sclera and proceed along the sclera or corneal regions, **Fig 1**. The length of the scarring response was quantified for its distance from the centre of the TM gutter using the calipers tool on Image J (https://imagej.net/ij) with the central corneal thickness used to calibrate the distance in microns, **Fig 1**.

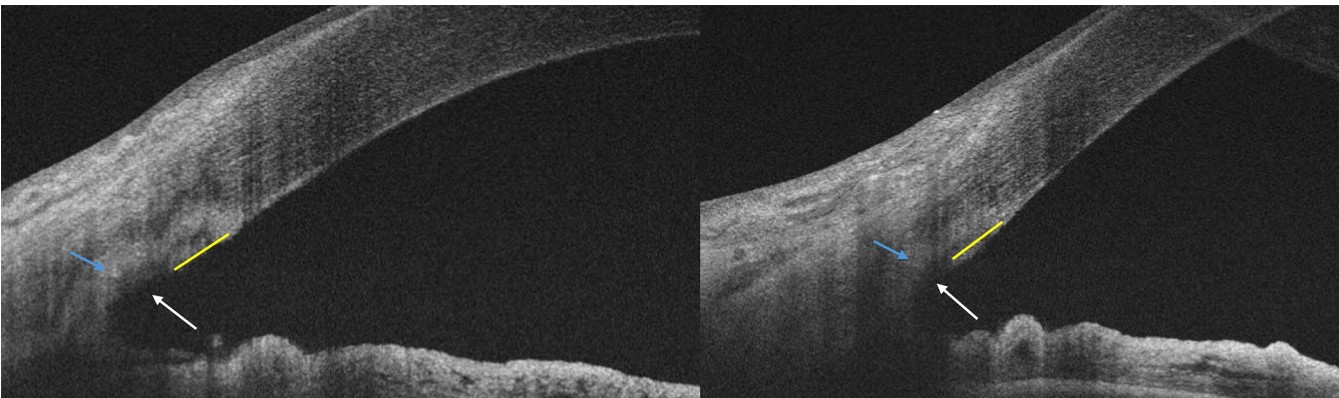

**Fig 1.** A and B show the method of measurement of the scarring response (yellow arrow) from the TM saucer/gutter (white arrow) measured on Image J and may be associated with scarring beneath the gutter (blue arrow)-See text for full description.

## Statistics

The analysis included examination of various factors such as the number of medications administered before and after surgery, clinical and demographic details, highest intraocular pressure (IOP) before surgery, intraoperative complications, follow-up IOP measurements during visits, during follow-up, and the necessity for medications or additional surgeries. Additionally, AS-OCT and gonioscopic findings such as peripheral anterior synechiae (PAS), adhesions, hypertrophic scar formation, or closure of the trabecular meshwork (TM) cleft were analysed for their correlation with postoperative IOP outcomes and surgical success. Statistical analysis was performed using Stats Corp (USA, version 13). Descriptive statistics were presented as standard deviation with mean deviation or median with interquartile range, and continuous variables were expressed as proportions. Comparisons between minimally invasive glaucoma surgery (MIGS) and Goniotomy-Assisted Transluminal Trabeculotomy (GATT) were conducted using unpaired Student's t-test, with statistical significance set at $p < 0.05$. AS-OCT predictors of surgical success or failure were determined through multivariate regression, with variables demonstrating $p < 0.02$ in univariate analysis considered for inclusion into the multivariate model.

# Results

Of 292 eyes that underwent MIT or GATT between March 2022- October 2023, 16 were excluded owing to poor quality scans. Of 274 eyes of 270 patients, we selected 73 eyes of 67 patients fulfilling inclusion criteria (M:F = 45:22) with $\geq$6 months of follow-up after MIT (n = 41) or GATT (n = 32) with or without combined cataract surgery. This included 49 POAG, 13 PACG, and 11 PXG eyes. Three patients underwent GATT in one eye and MIT in the other eye.

The preoperative IOP was high in both groups (28± 6.1mm Hg), with 28 eyes having a preoperative IOP>30mm Hg at the time of surgery. All eyes were on >/ = 2 medications, with 42 eyes being on >3 medicines at the time of surgery. The IOP reduced significantly in all eyes, with 18 eyes requiring one mediation for IOP control at a mean follow-up of 8±3.2 months. There was no difference between the final IOP between the eyes that underwent GATT (12 ±0.9mm Hg) or MIT (13±1.2mm Hg), p = 0.2. Intraoperative severe bleeding was seen in 2 eyes one of which underwent AC washout twice and was later found to be on anticoagulants which the patient was unaware of at the time of surgery. The other patient had a slow spontaneous recovery over 2.6 months with no risk factor found on thorough investigation with a slow dissolution of the clot in the AC and no recurrent bleeding. Postoperative macrohyphema was seen in the above 2 eyes with microhyphema seen in 12 eyes at day 1. None of the 71 eyes required additional interventions for hyphema that resolved spontaneously. One myope had a small cyclodialysis cleft which however did not result in hypotony and closed spontaneously at 2 months after surgery. No patient experienced iridodilalysis or Descemet's detachment (DMD) in this study.

IOP spikes were seen in 28 eyes (8 immediate spikes within 1 week of surgery and delayed spikes within 2 weeks- 2 months after surgery), of which 12 required medications for IOP control. Thirteen eyes were diagnosed as steroid responders, and the IOP was controlled with a switch to low-potent steroids while in 32 eyes, the IOP stabilized spontaneously at 1 month without the need for interventions or medications.

## AS-OCT parameters

Evaluating the healing at the TM region after either surgery, a hyperreflective scar was discernible in the region as early as 1 week after surgery. The response seemed to originate or be

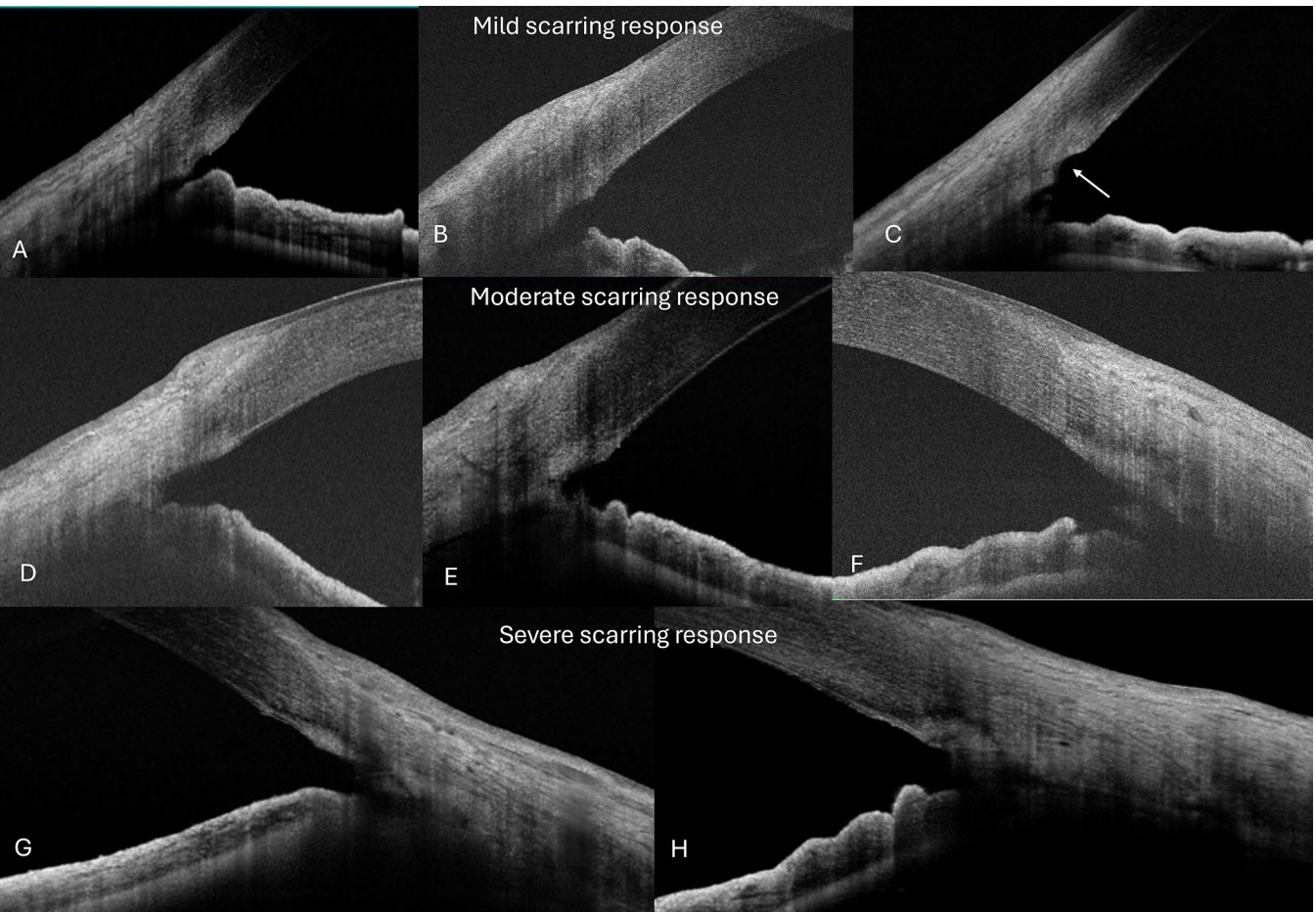

**Fig 2.** A-C shows mild scarring response after MIGS on ASOCT, D-F show moderate scarring response while G,H show severe scarring responses extending on to the sclera and cornea >500 microns-See text for full definition of scarring responses.

restricted to the sclera underneath the saucer or gutter in the region of the surgery, which then extended in the sclera and/or the cornea. There was no eye where the scarring or hyperreflective response was seen in the cornea area adjacent to the surgical site. While many eyes had extension of the scarring into the sclera, not all eyes had healing scars extending into the cornea. There was no remodelling seen after 3-4months after surgery. Based on the AS-OCT patterns of hyperreflectivity, the hypertrophic healing scar in the TM regions was classified as mild, moderate, and severe which were defined as below, **Fig 2**.

Nil: Nil or very minimally discernible hyperreflectivity in the TM region.

Mild-A hyperreflective scarred region (compared to the cornea) that extends from the region beneath the TM gutter to the cornea to a distance of ≤250 microns.

Moderate: A hyperreflective scarred region (compared to the cornea) that extends from the region beneath the TM gutter to the cornea to a distance of ≥250–500 microns.

Severe: A hyperreflective scarred region (compared to the cornea) that extends from the region beneath the TM gutter to the cornea to a distance of >500 microns.

Three patterns of healing responses were seen-healing underneath an open saucer or gutter at the TM region, healing underneath the incised TM with a filled-up saucer or gutter, and extensive scarring with closed gutter in the TM region and the adjacent cornea/sclera linearly in a filled-up trench pattern.

**Table 1. Clinical characteristics of patients that were evaluated using ASOCT after MIT or GATT.**

| Variable | Mean± SD or N |
|---|---|
| Age (years) | 62±12.7 |
| Gender (Male:Female) | 45:22 |
| Diagnosis | |
| POAG | 49 |
| PXG | 11 |
| PACG | 13 |
| Baseline IOP (mm Hg) | 28±6.1 |
| Medications | 3±1.2 |
| Mean deviation (dB) | -15±9.4 |
| IOP 1month (mm Hg) | 12±5.4 |
| IOP 3months (mm Hg) | 16±4.4 |
| IOP 6months (mm Hg) | 15±3.2 |
| IOP at final follow-up (months) | 8±3.2 |

PXG-Pseudoexfoliation glaucoma, PACG-primary angle closure glaucoma, POAG-Primary open angle glaucoma, IOP-intraocular pressure

The open gutter pattern was more common in eyes with MIT than GATT. The trench form of scarring was more common with GATT (56.2%) eyes than MIT eyes (31.3%), p = 0.08, **Table 1**, **Figs 3 and 4**. Twelve eyes with early IOP spikes in GATT, and 5 eyes after MIT had a trench form of scarring on AS-OCT and required 1–2 medicines for IOP control at final follow-up, **Table 2**.

There was a greater number of eyes showing severe scarring seen in GATT eyes (> 65%) than in MIT (31%), p = 0.003, **Table 2**, **Fig 4**. Mild scarring was seen in 24 eyes, moderate in 15 eyes, and severe in 34 eyes, **Table 2**. Yet, this did not reflect in the final IOP or number of medications though the complication rates were maximal in the MIT group than GATT with more IOP spikes, **Table 2**.

Postoperative PAS in 1–2 quadrants in the nasal incised region (apart from preexisting areas in PACG eyes) was seen in 28 of 73 eyes (12 MIT,16 GATT), with >2 quadrants seen in 6 eyes at 3 months after GATT. TM reattachment was seen in 10 eyes with GATT only, all with adjacent PAS in the same region, **Fig 4**.

Correlating the pattern or the extent of scarring with the final IOP or the need for medications, the eyes with a trench form of scarring response and extending > 500 microns (severe) from the TM gutter were the ones likely to need medications for IOP control. Independently, the scarring pattern and the pattern of scaring were not found to influence the final IOP or need for medications. However, the pattern or extent of scarring did not influence the overall final IOP or surgical outcome. Other clinical variables also did not influence the final IOP in MIT or GATT eyes.

## Discussion

This study found that the eyes with trench or linear form of severe scarring response at the TM region after MIT or GATT were more likely to need medicines for IOP control though it did not outcome the final surgical outcome or success. GATT eyes were more likely to have moderate-severe scarring than MIT, with more eyes having a trench pattern compared to open gutter/saucer pattern in MIT eyes as a healing response. These differential responses did not seem to impact the outcome or the final IOP in this study.

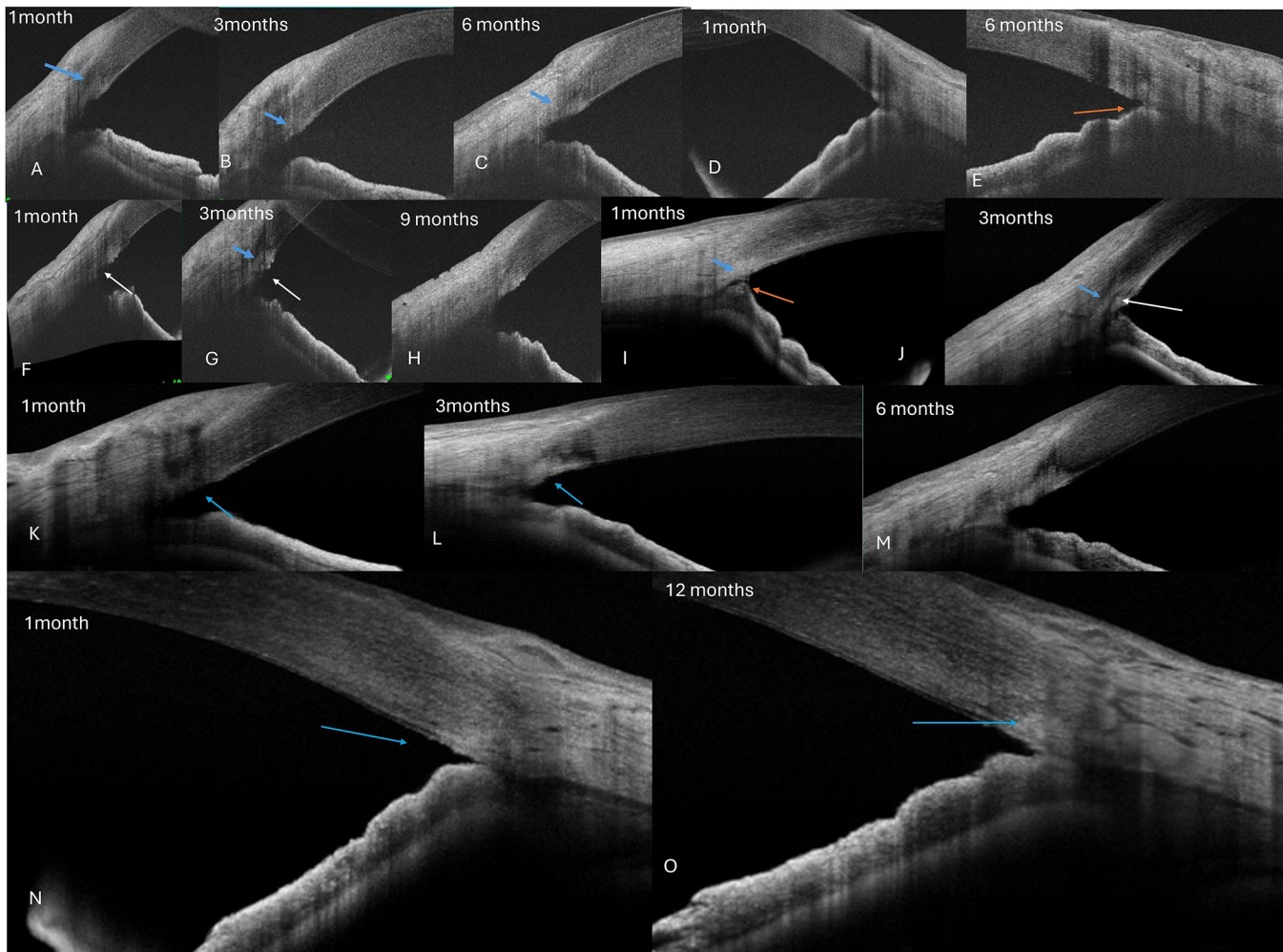

**Fig 3. Scarring responses on ASOCT after microincisional trabeculectomy.** A-C and F-H show mild scarring responses (blue arrow) that originate and are restricted to the region beneath the TM open gutter (white arrow) over 1->6months. D-E and I-J show similar healing responses with focal peripheral anterior synechiae (red arrow) and sub-TM scarring (blue arrow) and a closed gutter (white arrow). K-M show severe scarring in a trench pattern and a closed TM gutter at 3 months (L-blue arrow) that extends into the sclera and cornea (blue arrow). N and O show moderate scarring with a closed TM gutter at 1 month (blue arrow).

While MIT entails a gradual removal of the trabecular meshwork (TM) over 3–5 clock hours using microinstruments, GATT involves the insertion of a microcatheter or a 5–0 prolene suture into the SC, followed by tearing of the catheter or suture ends to rip away or disinsert the TM [5, 6, 9, 10]. Both procedures require meticulous precision and care to prevent inadvertent damage to the iris or cornea. MIT is characterized by a more controlled and gentle approach, devoid of forceful tearing, which may account for the lower incidence of complications compared to GATT [7]. It remains unclear whether this discrepancy contributes to the more frequent occurrence of trench-like scarring observed in GATT compared to MIT. An open saucer represents the open SC, which is GATT is guarded by the disinserted TM leaflet seen adjacent to the SC. Removal of the TM ensures that there is no TM reattachment in MIT as seen in this study. In contrast, 10 eyes with TM attachment after GATT also developed adjacent PAS in this study. This may imply that the removal of the disinserted TM (which is rendered dysfunctional) in GATT may prevent reattachment after GATT. It is still unclear

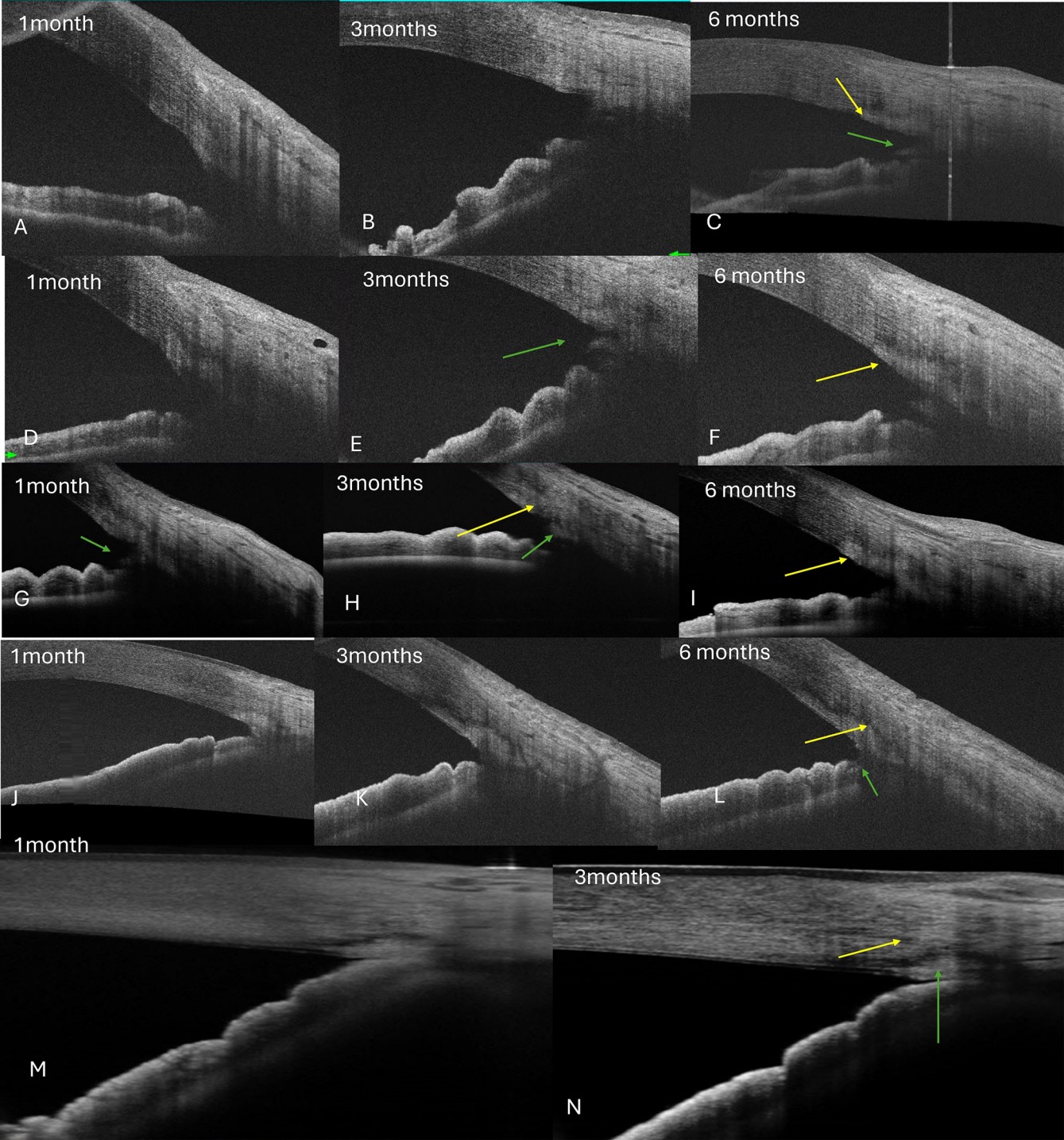

**Fig 4. Scarring responses on ASOCT after GATT.** A- shows an open saucer which then shows mild scarring at 3-6months (yellow arrow) with the open TM leaflet and SC (C-green arrow) that extends linearly onto the sclera at 6months. D-F and G-I show moderate scarring responses, with an open TM leaflet (green arrow) that gradually reattaches (H, I) and scarring extending into the cornea at 6 months (yellow arrow).J-L show severe scarring responses extending into the sclera and cornea in a trench pattern and TM reattachment (green arrow). M and N show similar healing responses (yellow arrow) with TM reattachment (green arrow).

**Table 2. Comparison of postoperative healing responses seen on ASOCT after MIT of GATT-see text for full description.**

|  | MIT (n = 41) | GATT (n = 32) | P value |
|---|---|---|---|
| PAS | 12 | 16 | 0.07 |
| Hyphema (total) | 3 | 11 | 0.003 |
| Transient | 2 | 10 |  |
| macrohyphema | 1 | 1 |  |
| IOP spikes (total) | 7 | 21 | 0.01 |
| Immediate | 2 | 8 |  |
| Early | 5 | 13 |  |
| Healing response |  |  |  |
| NIl | 0 | 0 |  |
| Mild | 20 | 4 | 0.001 |
| Moderate | 8 | 7 | 0.8 |
| Severe | 13 | 21 | 0.003 |
| Open Saucer/gutter | 18 | 10 | 0.2 |
| Closed saucer/gutter | 11 | 4 | 0.3 |
| Trench scar | 12 | 18 | 0.08 |
| Final medicines | 0.4±0.2 | 0.7±0.2 | 0.01 |
| TM reattachment | 0 | 10 | <0.001 |
| Miscellaneous | 1 cyclodialysis | 0 | 0.3 |
| Additional surgery/interventions | 1 AC WASH | 1 AC wash | 0.8 |

PAS-peripheral anterior synechiae, AC-anterior chamber, IOP-Intraocular pressure

whether this may affect the healing response postoperatively compared to eyes where the TM is left alone after GATT.

Wound healing in any area typically encompasses hemostasis, the influx of inflammatory cells, subsequent collagen production by nearby fibroblasts, and tissue remodelling [12–20]. While numerous investigations have scrutinized healing processes post-trabeculectomy, research on such responses in MIGS is scant [7, 12, 20]. In this study, the scarring response in the surgical site seemed to originate from the sclera underneath the open SC area or gutter/saucer. Schlemm's canal primarily contains blood vessels and endothelial cells [21–24]. It is a circular vessel encased in the sclera lined by endothelial cells and runs parallel to the corneal endothelium. While lymphatic vessels are not traditionally considered to be present within the SC, recent research has suggested the possibility of lymphatic vessels associated with the SC, particularly in the context of macromolecule clearance and immune responses [23, 24]. However, the exact nature and significance of lymphatic vessels within the SC are still areas of ongoing research and debate in the field of ophthalmology. This may provide the basis for understanding the wound healing responses after MIGS procedure and their impact on the surgical outcomes. This study evaluated the healing response at the angle using ASOCT. Yet, we have to recall the importance of the changes that may occur in the distal parts of the outflow pathway, namely the collector channels and the episcleral venous plexuses. Studying these systems is currently challenging and may be studied using other imaging modalities like OCT-angiography or aqueous angiography [21, 24]. This may also require animal studies for histo-pathological changes to be studied in different parts of the outflow pathway.

We understand that irreversible damage in glaucoma occurs due to the inability of the trabecular meshwork (TM) to regenerate once removed. While it is presumed that the anterior TM or the transition zone of the TM may harbor progenitor or stem cells, it remains unclear whether stem cells are recruited after MIGS [25]. The presence of fibroblastic scarring may

lead to the closure of the SC after any MIGS procedure. Interestingly, the presence of such scarring did not necessarily result in surgical failure in this study, but rather indicated the need for medication, particularly in cases where the scar was severe and extended onto the cornea in a trench-like pattern.

It is uncertain whether this scarring represents a purely fibroblastic response within the SC or an aborted/aberrant regenerative response from the adjacent TM with epithelial-mesenchymal transition similar to glial scarring observed after retinal injury [26]. Therefore, further studies are imperative to evaluate scarring after MIGS procedures in animal models. These studies should focus on understanding the wound healing responses and their origins to comprehend their implications fully.

This study had several limitations. This was a prospective study that compared only MIT and GATT and did not look at procedures like BANG, goniotomy or stents. We also studied the healing response in the nasal region for comparing MIT and GATT and did not study the other quadrants in eyes that underwent GATT. Also, we did not include the surgeries done by other surgeons since this may induce a bias due to differences in surgery patterns on the scarring responses. We also did not include secondary glaucoma for similar reasons. We did not evaluate the conjunctival changes including the scleral lake or the outflow pathway on AS-OCT nor did we measure the SC dimensions in this study since our objective was to study the healing responses. Nevertheless, we believe that scarring at the TM region after MIGS opens up new areas of research and possibilities to understand how the outflow pathway responds to injury. Further research in animal models may give insights into the origins of the fibroblastic response or their implications in understanding the pathogenesis of TM injury.

## Acknowledgments

Hyderabad Eye Research Foundation.

## Author Contributions

**Conceptualization:** Aparna Rao.

**Data curation:** Aparna Rao, Sujoy Mukherjee.

**Formal analysis:** Aparna Rao, Sujoy Mukherjee.

**Investigation:** Aparna Rao, Sujoy Mukherjee.

**Methodology:** Aparna Rao.

**Project administration:** Aparna Rao.

**Software:** Aparna Rao.

**Supervision:** Aparna Rao.

**Validation:** Aparna Rao.

**Visualization:** Aparna Rao.

**Writing – original draft:** Aparna Rao.

**Writing – review & editing:** Aparna Rao, Sujoy Mukherjee.

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
