## [Decision Letter · Decision Letter 0]

8 May 2024

PONE-D-24-13883Healing responses at the angle after micro-invasive glaucoma surgery-an AS-OCT studyPLOS ONE

Dear Dr. Rao,

Thank you for submitting your manuscript to PLOS ONE. After careful consideration, we feel that it has merit but does not fully meet PLOS ONE’s publication criteria as it currently stands. Therefore, we invite you to submit a revised version of the manuscript that addresses the points raised during the review process.

We look forward to receiving your revised manuscript.

Kind regards,

Shinji Kakihara, M.D.,Ph.D.

Academic Editor

PLOS ONE

Journal Requirements:

Additional Editor Comments:

Dear Authors,

The manuscript entitled “Healing responses at the angle after micro-invasive glaucoma surgery-an AS-OCT study” has been thoroughly reviewed. All reviewers have expressed significant interest in your study. However, they have also raised some important concerns. Therefore, we invite you to submit a revised version of the manuscript addressing the points mentioned during the review process.

Reviewers' comments:

Reviewer's Responses to Questions

**Comments to the Author**

1. Is the manuscript technically sound, and do the data support the conclusions?

Reviewer #1: Partly

Reviewer #2: Yes

2. Has the statistical analysis been performed appropriately and rigorously? 

Reviewer #1: Yes

Reviewer #2: Yes

3. Have the authors made all data underlying the findings in their manuscript fully available?

Reviewer #1: Yes

Reviewer #2: Yes

4. Is the manuscript presented in an intelligible fashion and written in standard English?

Reviewer #1: No

Reviewer #2: Yes

5. Review Comments to the Author

Reviewer #1: I have read this paper with great interest and congratulate authors for their innovative view on this specific topic. As a MIGS surgeon with higher load of surgical glaucoma patients, I have had a considerable amount of experience with GATT. Since, this surgery does not necessarily needs higher costs with no reimbursement issues in a typical developing country. As for this particular paper, I actually hesitate whether these reported OCT findings might have relevance on wound healing response after angle based MIGS, as authors also could not generate certain and satisfactory statistical findings to support their final conclusion.

I have been considering this so called wound healing response soon after beginning of GATT which I observed several diverse findings on OCT imaging. Unfortunately, OCT could allow analysing to a certain extent; namely, we are not able to see beyond proximal outflow portion at least during routine clinical practice.

If you initially aim to analyse healing response in the angle at a broader extent, here, in my opinion, the impact of collector system wound healing response also may have relevance on surgical outcomes, more likely on IOP spikes (just like an IOP spike after a typical closure of cyclodialysis cleft). What I actually mean is collector morphology should be assessed further in order to reach a more definitive consequence. This includes that hinge morphology, collector entrance and perhaps finally collector orifis damage. Today’s OCT technology unfortunately does not allow this assessment at least in the clinic. Maybe, you should reorganize your title and purpose.

Yet, you exhibit no influence of your OCT findings on overall success, but you stated about severe scarring may end up with reduced medications. But, I can see no robust analysis to satisfy the readers about this finding except for a simple intergroup comparison in Table 2. I mean this conclusion remains in a perceivable level.

Your main finding is about the presence of scarring morphology on OCT and its influence on medication use. What about the effect of distal outflow system on clinical outcomes. I found no words at least in the discussion.

I totally agree with you regarding an animal study in order to search for a histopathological clue in outflow healing response after angle surgery.

Amount of PAS has already been shown not to affect IOP levels in open angle cases (you can see; Gunay M, et al. Evaluation of peripheral anterior synechia formation following gonioscopy-assisted transluminal trabeculotomy surgery. Int Ophthalmol.)

I believe you should exclude those cases with angle closure.

I like the way you hypothesize some concerns in discussion, but again I should reiterate the possible effect of collector system alterations after such surgical techniques.

Table 2 ==> “es seen”

Paper may be reviewed again by the authors because of possible syntax errors.

Reviewer #2: I had the pleasure to review the manuscript by Aparna Rao describing structural alterations and healing responses in the trabecular meshwork region with optical coherence tomography following after gonioscopy

assisted transluminal trabeculotomy and MIT. The manuscript is well written and fluent. Some corrections need to be addressed :

abstract: methods: peripheral anterior synechiae, PAS, brackets should be for the acronym

abstract: methods: (microincisional trabeculectomy) MIT. brackets should be for the acronym

abstract: methods: TM gutter... explain TM

the aim of the study is clear and well defined. The methods are well developed and explained. The Statistical analysis is well performed and the results are solid. The discussion is well developed.

Given the prospective nature of the study it would have been interesting to have preoperative scans in order to define a real baseline image to follow up.

6. PLOS authors have the option to publish the peer review history of their article (what does this mean?). If published, this will include your full peer review and any attached files.

Reviewer #1: No

Reviewer #2: No

---

## [Author Response · Author response to Decision Letter 0]

14 May 2024

¬¬To,

The Editor,

Dear Sir/Madam,

We hereby submit our revised manuscript “Healing responses at the angle after micro-invasive glaucoma surgery-an AS-OCT study.“for review for publication in your journal. We are also enclosing the reply to reviewer’s comments with point-point clarifications to the suggestions. We are thankful and delighted with the suggestions of the reviewers and have incorporated all suggestions into the text that has helped improve the impact of the study. We would welcome any more suggestions or queries which can further improve our manuscript.

All the authors have contributed equally towards the preparation of the manuscript and have no financial or proprietary interest in the products used in the study. We also declare that this article has not been published previously or is under review with any other journal.

a) All acknowledgments and financial disclosures/funding information is included in the manuscript. 

b) All data have been given in the manuscript with additional patient-identifying information that may be shared after consent upon request. 

Thanking you

The manuscript entitled “Healing responses at the angle after micro-invasive glaucoma surgery-an AS-OCT study” has been thoroughly reviewed. All reviewers have expressed significant interest in your study. However, they have also raised some important concerns. Therefore, we invite you to submit a revised version of the manuscript addressing the points mentioned during the review process.

Answer: We have now incorporated all suggestions of the reviewers and have modified the manuscript as per their suggestions. We would welcome further suggestions if any to improve the manuscript further.

Reviewers' comments:

5. Review Comments to the Author

Reviewer #1: I have read this paper with great interest and congratulate authors for their innovative view on this specific topic. As a MIGS surgeon with higher load of surgical glaucoma patients, I have had a considerable amount of experience with GATT. Since, this surgery does not necessarily needs higher costs with no reimbursement issues in a typical developing country. As for this particular paper, I actually hesitate whether these reported OCT findings might have relevance on wound healing response after angle based MIGS, as authors also could not generate certain and satisfactory statistical findings to support their final conclusion.

Answers: We thank the reviewer for the encouragement on this topic. We have presented the details and findings in a very explicit and honest manner, and we agree, this paper like any other scientific study opens up more questions than answers. Yet, we believe this paper tries to look into some aspects of wound healing in the angle after MIGS which definitely needs further in-depth studies. 

I have been considering this so-called wound healing response soon after beginning of GATT which I observed several diverse findings on OCT imaging. Unfortunately, OCT could allow analysing to a certain extent; namely, we are not able to see beyond proximal outflow portion at least during routine clinical practice.

Answer: Yes, we agree with the reviewer that ASOCT allows visualisation of some parts of the outflow system only barring the collector channels or the episcleral plexus, which will require alternate imaging methods like aqueous angiography or OCT-angiography. Our study only focussed on ASOCT to evaluate the imaging responses at the angle only using ASOCT.

If you initially aim to analyse healing response in the angle at a broader extent, here, in my opinion, the impact of collector system wound healing response also may have relevance on surgical outcomes, more likely on IOP spikes (just like an IOP spike after a typical closure of cyclodialysis cleft). What I actually mean is collector morphology should be assessed further in order to reach a more definitive consequence. This includes that hinge morphology, collector entrance and perhaps finally collector orifis damage. Today’s OCT technology unfortunately does not allow this assessment at least in the clinic. Maybe, you should reorganize your title and purpose.

Answer: Yes, we agree, ASOCT only allows imaging of the angle per se and not the entire collector system responses and recruitment after MIGS which may be evaluated better with OCT-A or aqueous angiography. The title also therefore only mentions :angle” keeping in mind this limitation. 

Yet, you exhibit no influence of your OCT findings on overall success, but you stated about severe scarring may end up with reduced medications. But, I can see no robust analysis to satisfy the readers about this finding except for a simple intergroup comparison in Table 2. I mean this conclusion remains in a perceivable level.

Answer: We still do not understand the relevance of ASOCT findings on the overall clinical success rates since the latter would depend on the responses of the other parts of the outflow pathway namely the collector channels and the venous plexuses. We have only studied the ASOCT findings in the angle after MIGS and do not believe that these findings can correlate directly with the success rates and this is exactly why we believe we did not see such correlation in this study too. However, since this is pertinent point, we have included the same in the discussion in the revised manuscript. 

Your main finding is about the presence of scarring morphology on OCT and its influence on medication use. What about the effect of distal outflow system on clinical outcomes. I found no words at least in the discussion.

I totally agree with you regarding an animal study in order to search for a histopathological clue in outflow healing response after angle surgery.

Answer: Yes, we agree with the reviewer that the other parts of the outflow pathway and their responses after MIGS hold utmost importance to understand the effect of MIGS on IOP. Animal studies can help give histopathological changes seen in the outflow pathway after MIGS which will add valuable information on the wound healing responses. Yet, we currently do not have an easy way of evaluating these systems in-vivo in humans intraoperatively or postoperatively; Aqueous angiography and OCT-A (not easily available at all centres) give some information on the collector channel recruitment but the pressure differences in the collector channels or venous plexuses caused after MIGS needs animal studies or doppler which mare nor readily available. The pressure adjustments in the collector channels after MIGS or their collapse is only presumptive and needs some robust imaging method like doppler in the future studies. This study only focussed on the scarring responses in the angle using ASOCT after MIGS. This is only one piece in the whole jigsaw puzzle of the physiology of the outflow pathway. We have now included this point in ths discussion and thank the reviewer for this suggestion since this is very relevant and pertinent. 

Amount of PAS has already been shown not to affect IOP levels in open angle cases (you can see; Gunay M, et al. Evaluation of peripheral anterior synechia formation following gonioscopy-assisted transluminal trabeculotomy surgery. Int Ophthalmol.)

I believe you should exclude those cases with angle closure.

Answer: Yes, it is surprising that PAS did not impact the success rates in MIGA as shown in other studies and in this study. We did not exclude these cases since the PAS we refer to are not PAS that were pre-existing but rather those that developed postoperatively and the procedure performed was the same in PACG eyes too in areas away from the PAS. We have not seen clinical differences in outcomes after MIT or GATT in POAG and PACG in our earlier studies which is why we included PACG eyes as well in this study. 

I like the way you hypothesize some concerns in discussion, but again I should reiterate the possible effect of collector system alterations after such surgical techniques.

Table 2 ==> “es seen”

Paper may be reviewed again by the authors because of possible syntax errors.

Reviewer #2: I had the pleasure to review the manuscript by Aparna Rao describing structural alterations and healing responses in the trabecular meshwork region with optical coherence tomography following after gonioscopy

assisted transluminal trabeculotomy and MIT. The manuscript is well written and fluent. Some corrections need to be addressed :

abstract: methods: peripheral anterior synechiae, PAS, brackets should be for the acronym

abstract: methods: (microincisional trabeculectomy) MIT. brackets should be for the acronym

abstract: methods: TM gutter... explain TM

Answer: We thank the reviewer for the encouraging and supportive words, and we have made changes in the revised manuscript as suggested by the reviewer. 

the aim of the study is clear and well defined. The methods are well developed and explained. The Statistical analysis is well performed, and the results are solid. The discussion is well developed.

Given the prospective nature of the study it would have been interesting to have preoperative scans in order to define a real baseline image to follow up.

Answer: Yes, the results of this study has prompted us to investigate further. We are now doing preoperative scans of the other eye which later underwent surgery, and the findings are similar to this study. The results of the new study would be shared when ready with the results with the reviewer definitely. The prospective study looks at clinical outcomes also prospectively which also would be shared when ready. ________________________________________

---

## [Decision Letter · Decision Letter 1]

5 Jun 2024

Healing responses at the angle after micro-invasive glaucoma surgery-an AS-OCT study

PONE-D-24-13883R1

Dear Dr. Rao,

We’re pleased to inform you that your manuscript has been judged scientifically suitable for publication and will be formally accepted for publication once it meets all outstanding technical requirements.

Kind regards,

Shinji Kakihara, M.D.,Ph.D.

Academic Editor

PLOS ONE

Additional Editor Comments (optional):

The authors have adequately addressed the reviewers' comments. I believe that the current manuscript is suitable for publication in PLOS ONE.

Reviewers' comments:

Reviewer's Responses to Questions

**Comments to the Author**

1. If the authors have adequately addressed your comments raised in a previous round of review and you feel that this manuscript is now acceptable for publication, you may indicate that here to bypass the “Comments to the Author” section, enter your conflict of interest statement in the “Confidential to Editor” section, and submit your "Accept" recommendation.

Reviewer #1: All comments have been addressed

Reviewer #2: All comments have been addressed

2. Is the manuscript technically sound, and do the data support the conclusions?

Reviewer #1: Yes

Reviewer #2: Yes

3. Has the statistical analysis been performed appropriately and rigorously? 

Reviewer #1: Yes

Reviewer #2: Yes

4. Have the authors made all data underlying the findings in their manuscript fully available?

Reviewer #1: (No Response)

Reviewer #2: Yes

5. Is the manuscript presented in an intelligible fashion and written in standard English?

Reviewer #1: Yes

Reviewer #2: Yes

6. Review Comments to the Author

Reviewer #1: Appreciate authors for revision process of this paper. I still believe that the detailed analysis of distal outflow regarding its wound healing response requires comprehensive assessment.

Reviewer #2: I have reviewed the manuscript and the authors have modified the manuscript. The paper in my opinion offers an innovative point of view and it is well described and defined.Despite the lack of preoperative OCT finding and the limitiation of our current OCT technologies, the image analysis can offer views for further studies.

7. PLOS authors have the option to publish the peer review history of their article (what does this mean?). If published, this will include your full peer review and any attached files.

Reviewer #1: No

Reviewer #2: No

---

## [Editor Report · Acceptance letter]

17 Jun 2024

PONE-D-24-13883R1 

PLOS ONE

Dear Dr. Rao, 

I'm pleased to inform you that your manuscript has been deemed suitable for publication in PLOS ONE. Congratulations! Your manuscript is now being handed over to our production team.

Kind regards, 

on behalf of

Dr. Shinji Kakihara 

Academic Editor

PLOS ONE